# Bacterial Production of CDKL5 Catalytic Domain: Insights in Aggregation, Internal Translation and Phosphorylation Patterns

**DOI:** 10.3390/ijms25168891

**Published:** 2024-08-15

**Authors:** Andrea Colarusso, Concetta Lauro, Luisa Canè, Flora Cozzolino, Maria Luisa Tutino

**Affiliations:** 1Department of Chemical Sciences, University of Naples Federico II, Complesso Universitario Monte S. Angelo, Via Cintia 4, 80126 Naples, Italy; and.colarusso@gmail.com (A.C.); concetta.lauro@unina.it (C.L.); flora.cozzolino@unina.it (F.C.); 2CEINGE Advanced Biotechnologies, Via G. Salvatore 486, 80145 Naples, Italy; canel@ceinge.unina.it; 3Department of Translational Medical Sciences, University of Naples Federico II, Via Sergio Pansini 5, 80131 Naples, Italy; 4Istituto Nazionale Biostrutture e Biosistemi I.N.B.B., Viale Medaglie D’Oro 305, 00136 Roma, Italy

**Keywords:** CDKL5, kinase, phosphorylation, aggregation, TXY motif, chaperones, solubility tag

## Abstract

Cyclin-dependent kinase-like 5 (CDKL5) is a serine/threonine protein kinase involved in human brain development and functioning. Mutations in CDKL5, especially in its catalytic domain, cause a severe developmental condition named CDKL5 deficiency disorder. Nevertheless, molecular studies investigating the structural consequences of such mutations are still missing. The CDKL5 catalytic domain harbors different sites of post-translational modification, such as phosphorylations, but their role in catalytic activity, protein folding, and stability has not been entirely investigated. With this work, we describe the expression pattern of the CDKL5 catalytic domain in *Escherichia coli* demonstrating that it predominantly aggregates. However, the use of solubility tags, the lowering of the expression temperature, the manual codon optimization to overcome an internal translational start, and the incubation of the protein with K^+^ and MgATP allow the collection of a soluble catalytically active kinase. Interestingly, the resulting protein exhibits hypophosphorylation compared to its eukaryotic counterpart, proving that bacteria are a useful tool to achieve almost unmodified CDKL5. Posing questions about the CDKL5 autoactivation mechanism and the determinants for its stability, this research provides a valuable platform for comparative biophysical studies between bacterial and eukaryotic-expressed proteins, contributing to our understanding of neurodevelopmental disorders associated with CDKL5 dysfunction.

## 1. Introduction

Cyclin-dependent kinases-like 1-5 (CDKL1-5) are underexplored protein kinases belonging to the CMGC clade, a vast group of proteins including cyclin-dependent kinases (CDKs), mitogen-activated kinases (MAPKs), glycogen synthase kinases (GSKs) and CDKLs [1]. Although CDKLs share similar sequences to CDKs, some amino acid substitutions make them unlikely to be cyclin binders [2]. Furthermore, the conservation of a TXY motif in their catalytic domain underlies an activation mechanism similar to MAPKs, which generally require the double phosphorylation of both T and Y residues for full activation [3,4]. These same features are shared by another ancient family of poorly studied kinases, the ros cross-hybridizing kinases (RCKs) that include MAK and MOK [5]. Besides the analogies in their catalytic domains, all CDKLs and RCKs also have the same modular architecture made of an N-terminal conserved catalytic domain and a C-terminal tail, which is predicted to be disordered [1].

Functionally speaking, little is known about CDKL1-4 but an overlap of CDKL5 with RCKs is evident given that all these three proteins intervene in cilia biology [2,5], suggesting even common regulatory mechanisms for their activities in model organisms [6,7,8]. Intriguingly, the RCK kinase MAK has been recently proven to phosphorylate some CDKL5 substrates [9]. Interestingly, CDKL5 is probably the most studied among such “dark kinases” (i.e., poorly understood kinases) [10,11] mainly due to the relevance of this protein in neuronal development and its involvement in an infantile neurodevelopmental disorder named CDKL5 deficiency disorder (CDD) [12,13]. The literature highlighted that the *CDKL5* gene is expressed in several tissues but is particularly crucial for brain development where its expression increases postnatally and stabilizes through adulthood [14,15,16]. Cellular studies showed that thanks to its disordered tail, CDKL5 localizes in different cellular districts [14]: the nucleus, where it regulates RNA splicing in nuclear speckles [17] and transcription at the level of double-strand DNA breaks [18]; synapses, where it modulates endocytic processes and synaptic stability [15,19,20,21]; dendrites, where it regulates dendritic spine morphology and the formation of focal adhesions [15,19,21]; and cilia, where it controls their average length [2].

CDD is predominantly caused by missense mutations in the catalytic domain of CDKL5 [12] but none of them has been mechanistically explored: most studies are based on either in vitro activity assays or cellular experiments that generally demonstrate a variable hypoactivity of each mutant [22,23,24,25]. However, the links between such hypoactivity and CDD complex symptomatology are still under investigation [26].

To overcome this limitation, a valid strategy for the production and purification of the CDKL5 catalytic domain (CDKL5ΔC) in bacteria is desirable. This would allow for a better structural characterization of such a domain, an understanding of the architectural consequences of CDD mutations, and opening the screening for drugs to stabilize CDD mutants.

In this work, we present a series of approaches adopted to achieve a purifiable form of CDKL5ΔC in *E. coli* strains taking into account genetic aspects (codon composition and an internal translational start), expression variables to achieve a soluble product (truncation boundaries, co-expression with chaperones, and fusion with solubility tags), and the levels of autophosphorylation of the protein activation loop. We demonstrate that CDKL5ΔC is mainly insoluble in bacteria and that the application of a previously adopted strategy (i.e., the overexpression of bacterial chaperones [27,28,29]) only stabilizes soluble inactive aggregates. The fusion of the target protein with solubility tags, the production at low temperatures, and the incubation of the protein with K^+^ and MgATP during purification allow for achieving a pure active kinase, instead. However, our analysis shows that the activation loop of the *E. coli*-produced CDKL5ΔC is significantly less phosphorylated than in the same protein achieved from Sf9 insect cells. This feature makes again CDKL5 similar to MAK [30] and suggests that further studies are needed to understand the CDKL5 activation mechanism and the influence of its phosphorylation on its overall stability (and vice versa).

## 2. Results

### 2.1. Evaluation of Multiple Variables for the Recombinant Production of CDKL5ΔC in E. coli

To define the critical parameters affecting the production of CDKL5ΔC in *E. coli*, we screened different variables (Table 1). The choice of kinase domain boundaries and the phosphorylation levels of the domain itself can determine the success of recombinant expression [31]. For these reasons, we selected three differently C-terminally truncated constructs based on the CDKL5 literature and commercial sources and produced the protein both as wt and kinase-dead (KD) mutant through the KK42,43RR double mutation (Figure 1). We tested different sequences whose codon composition has been optimized with different criteria and used different approaches (i.e., either a missense M10V mutation or a silent mutation) to stem an internal translation start which is typical for CDKL5 in bacteria [25]. Finally, given the propensity of eukaryotic kinases to partition into the insoluble fraction of bacterial lysates [32], we adopted different conventional strategies (i.e., modulation of expression temperature, N-terminal fusion with known solubility tags, and co-expression with bacterial protein chaperones) to stem this issue. However, we also tried to harvest active CDKL5ΔC from insoluble fractions (Table 1). For most of these trials, we used four readouts: yield and purity after protein isolation, level of phosphorylation of the CDKL5 activation loop (i.e., the TEY motif in the 169–171 amino acid stretch [3]), and activity against a known CDKL5 substrate, EB2 [22,23].

### 2.2. Influence of Codon Composition and Sumo Tag on E. coli Expression of CDKL5ΔC

The main examples about the recombinant expression of either full-length CDKL5 (flCDKL5) or its catalytic domain (CDKL5ΔC) involve a 1-303 construct purified from insect cells for its successful crystallization [2,33], a 1-352 construct expressed in *E. coli* BL21(DE3) [27] and a full-length Sumo-tagged variant expressed in the psychrophilic bacterium *Pseudoalteromonas haloplanktis* TAC125 (*P. haloplanktis* TAC125) [25]. Furthermore, the last paper reported that an M10V mutation was needed to avoid an internal translation start and achieve the product with an intact N-terminal extremity [25]. Based on these data, we decided to produce a CDKL5(1-352) variant with a C-terminal His-tag to replicate Kameshita’s construct, which already proved to be useful for the *E. coli* expression of CDKL5ΔC [27]. However, we kept the M10V mutation to avoid translational issues in the first place. Hence, this construct was named CDKL5(1-352)-His M10V.

As Kameshita et al. [27] needed the co-expression of GroELS with CDKL5ΔC in *E. coli*, we hypothesized that insolubility issues could arise during the overexpression of the kinase in this bacterium. For this reason, both a Sumo-tagged and an untagged construct were produced using the *P. haloplanktis* TAC125 codon-optimized gene as a template (Ph-labeled variants) [25]. To check the possible influence of the codon composition on the expression levels, an *E. coli* codon-optimized CDKL5(1-352) gene was also generated using a pET28-based plasmid as a template (WO2018005617A2 patent, Ec labeled variant). The three proteins were named Ph_Sumo-Tat-CDKL5(1-352)-His M10V (MW~57 kDa), Ph_Tat-CDKL5(1-352)-His M10V (MW~45 kDa), and Ec_Tat-CDKL5(1-352)-His (MW~44 kDa). All protein variants harbor an N-terminal Tat peptide for intracellular vehiculation in eukaryotic systems [34].

Protein production was evaluated in three *E. coli* strains: BL21(DE3), ArcticExpress(DE3) for Cpn10 and Cpn60 chaperones expression, and Shuffle T7 for cytoplasmic DsbC expression.

As all three constructs were insoluble at 37 °C, only the results achieved at 15 °C are reported. As visible in Figure 2, Ph_Tat-CDKL5(1-352)-His M10V was poorly produced in all the tested *E. coli* strains and its expression was only visible in an anti-His Western blot (Figure 2B, lanes *a*). On the other hand, Ph_Sumo-Tat-CDKL5(1-352)-His M10V (lanes *b* in the Figure 2) and Ec_Tat-CDKL5(1-352)-His (lanes *c* in the Figure 2) were equally expressed in BL21(DE3) and Shuffle T7 strains. Nevertheless, Ph_Sumo-Tat-CDKL5(1-352)-His M10V expression levels were significantly lowered in ArcticExpress(DE3) in comparison with the *E. coli* optimized variant.

As Ph_Tat-CDKL5(1-352)-His M10V was poorly expressed, it was not used in the rest of this work. On the other hand, Ec_Tat-CDKL5(1-352)-His (lanes *a* in Figure 3) and Ph_Sumo-Tat-CDKL5(1-352)-His M10V (lanes *b* in Figure 3) solubility levels were evaluated in the three strains at 15 °C. Ec_Tat-CDKL5(1-352)-His was totally insoluble in all three strains (lanes *in* for insoluble extracts and lanes *s* for respective soluble fractions in Figure 3). On the other hand, Ph_Sumo-Tat-CDKL5(1-352)-His M10V was partially soluble in the tested strains (Figure 3). Interestingly, while in BL21(DE3) and Shuffle T7 Ph_Sumo-Tat-CDKL5(1-352)-His M10V was mainly insoluble, it was totally absent in ArcticExpress(DE3) insoluble extract (Figure 3). Hence, even if this CDKL5ΔC variant was less expressed in this strain, it was entirely soluble. Based on these results, the Sumo-tagged CDKL5ΔC variant was used to perform purifications in native conditions.

### 2.3. Purification of Ph_Sumo-Tat-CDKL5(1-352)-His M10V from BL21(DE3) and ArcticExpress(DE3) Strains in Native Conditions 

In order to define if the increased solubility of Ph_Sumo-Tat-CDKL5(1-352)-His M10V when expressed in ArcticExpress(DE3) cells (Figure 3) could guarantee a higher protein recovery, IMACs carried out on BL21(DE3) and ArcticExpress(DE3) soluble lysates were compared. The total amount of CDKL5ΔC recovered from the two strains was similar (E1 and E2 fractions in Figure 4), but the contamination profile was different. In both cases, a major contaminant of ~60 kDa was mainly eluted with an intermediate wash step (W fractions in Figure 4) and partially contaminated final CDKL5ΔC preparations (E1 and E2). However, such a contaminant was more abundant in the ArcticExpress(DE3) derived fractions. The main difference between the two used strains is the expression of Cpn60 and Cpn10 in the ArcticExpress(DE3) strain. This observation suggests that the main CDKL5ΔC contaminant is Cpn60 when the purification is carried out from ArcticExpress(DE3) lysates, and its homolog Hsp60 when BL21(DE3) strain is used. Based on this result, ArcticExpress(DE3) was not used for preparative chromatographies of Ph_Sumo-Tat-CDKL5(1-352)-His M10V.

### 2.4. Effect of the Co-Expression of E. coli Molecular Chaperons on CDKL5ΔC Solubility 

CDKL5ΔC is mainly insoluble after overexpression in *E. coli* and our attempts to recover an active product from insoluble fractions failed (Appendix A). As CDKL5ΔC probably tends to copurify with Hsp60 and Cpn60 when expressed in BL21(DE3) and ArcticExpress(DE3), respectively (Figure 4), this may be indicative of folding issues that can be solved by overexpressing *E. coli* autologous molecular chaperones, as already proposed for CDKL5ΔC [26]. For this reason, we used the Takara plasmid set to co-express different combinations of chaperones with CDKL5ΔC variants. Given that depending on the protein size and structure, different chaperones can be recruited, differently C-terminally truncated CDKL5 variants were tested: 1-303, 1-352, and 1-498. For the same reason, they were produced in three different configurations: untagged, with a Sumo tag, and with a GST tag (Table 1). The choice of the truncation points depends on either literature reports or commercially produced CDKL5 variants (Figure 1). For brevity, only the results involving Sumo-tagged constructs are shown because they generated the most promising outcomes. Furthermore, the constructs with variable lengths gave rise to similar results (see Appendix A). Hence, only the data concerning Ph_Sumo-Tat-CDKL5(1-352)-His M10V are reported for the sake of uniformity with the experiments described in the main text.

Figure 5 shows the soluble fractions achieved after cellular lysis of BL21(DE3) co-expressing different combinations of chaperones with Ph_Sumo-Tat-CDKL5(1-352)-His M10V at 15 °C. 

The strain expressing Ph_Sumo-Tat-CDKL5(1-352)-His M10V alone (lane 5 in Figure 5) produced considerably less soluble protein in comparison with all the other strains overexpressing the molecular chaperones. In particular, the overproduction of the trigger factor (Tf) either alone or in combination with GroELS provided the best results (lanes 2 and 4, respectively, in Figure 5). On the other hand, DnaJ, DnaK, and GrpE had a cumulative effect with GroELS, as visible in lanes 6, 7, and 8.

The co-expression of chaperones with either untagged CDKL5ΔC or at higher temperatures did not give rise to better solubility (Appendix A).

### 2.5. Effect of the Co-Expression of E. coli Molecular Chaperons on CDKL5ΔC Activity 

Explorative IMACs were carried out on soluble lysates of *E. coli* strains expressing Ph_Sumo-Tat-CDKL5(1-352)-His M10V with or without molecular chaperones. As indicated in Table 2, chaperones allowed for increased recovery of the target protein. Nevertheless, most of it did not bind the Ni^2+^ resin, which is indicative of the fact that most of the protein made soluble by the chaperones is likely to be in an aggregated state. 

Purified CDKL5ΔC samples were then tested for their activity on EB2, a known CDKL5 substrate [22,23]. As visible in Figure 6, even if similar amounts of enzymes and substrate were used in each assay (Figure 6A), CDKL5ΔC purified from an *E. coli* strain not overexpressing any chaperone was considerably more active than the same protein purified from strains co-expressing different combinations of GroELS, trigger factor, and GrpE, DnaK, and DnaJ (Figure 6B). This result confirms further that overexpression of *E. coli* chaperones decreases the quality of CDKL5ΔC preparations, possibly through the stabilization of catalytically inactive folding intermediates.

### 2.6. Analysis of CDKL5ΔC Autophosphorylation by Western Blotting

In an attempt to recover CDKL5ΔC activity, we tested whether the enzyme’s activation loop could be trans-phosphorylated by catalytically active CDKL5. In this work, we have shown that CDKL5ΔC is phosphorylated on Y171 when purified in native conditions from soluble bacterial lysates, while it is not phosphorylated when it segregates in the insoluble fraction (Appendix A). To date, the mechanism of CDKL5 autophosphorylation is not entirely clear. It is not known if CDKL5 phosphorylates both T169 and Y171 and we do not know if such residues are also substrates of upstream kinases as in the case of many MAP kinases. Furthermore, conflicting papers report that CDKL5 is either intra-molecularly [3] or inter-molecularly autophosphorylated [4]. 

To shed light on these aspects, we analyzed T169 and Y171 phosphorylation states of a catalytically active CDKL5 variant (Ph_Sumo-Tat-CDKL5(1-498)-His M10V) and a shorter kinase-dead mutant (Ph_Sumo-Tat-CDKL5(1-352)-His M10V_KD) after incubation with MgATP both alone and mixed. After a reaction of 2 h at 30 °C, the reaction was stopped and the phosphorylation state of the proteins was evaluated by Western blot using two affinity-purified antibodies [23]. The catalytically active CDKL5ΔC variant was reactive to both anti-pTyr171 and anti-pThr169/pTyr171 antibodies, suggesting that the enzyme putatively autophosphorylates both residues (Figure 7). Phosphorylation levels did not change over time, though. On the other hand, the KD mutant was not reactive to pTyr171 and weakly reactive to anti-pThr169/pTyr171 antibodies, probably due to unspecific binding. Additionally, the KD mutant was not phosphorylated by Ph_Sumo-Tat-CDKL5(1-498)-His M10V, thereby suggesting that CDKL5 cis-phosphorylates its activation loop.

To be sure that this outcome was not a consequence of the quality of CDKL5ΔC preparations, the same assay was repeated by using a commercial CDKL5ΔC variant purified from Sf9 cells (#ab131695, Abcam, Cambridge, UK). Also, in this case, no clear trans-phosphorylation was visible (Appendix A). However, CDKL5ΔC isolated from insect cells showed so strikingly higher phosphorylation levels both in T169 and Y171 that the phosphorylation signals of the bacterially isolated CDKL5ΔC were obscured when analyzed on the same Western blot. Furthermore, unlike our preparations from *E. coli*, the enzyme obtained from the eukaryotic strain could increase its autophosphorylation over time when incubated with MgATP (Appendix A).

### 2.7. Analysis of CDKL5ΔC Autophosphorylation by Mass Spectrometry

Given that the specificity of the used anti-pTyr171 antibody has been demonstrated in the past [23] but the use of the anti-pThr169/pTyr171 from the same source (University of Dundee) has never been reported before, we wanted to confirm the results obtained by the Western blot analyses, by developing a suitable protocol of in gel trypsin hydrolysis and peptide analysis by mass spectrometry techniques on purified CDKL5ΔC from bacterial and insect cells (see Appendix A). Surprisingly, in both cases, we could only detect Y171 phosphorylation and no T169 phosphorylation. Furthermore, only 0.8% of the *E. coli*-produced enzyme was tyrosine phosphorylated while 20% of Sf9 cell-generated CDKL5ΔC was phosphorylated on the same residue in agreement with the intensity of Western blot signals. This result demonstrates that the anti-pThr169/pTyr171 antibody cannot distinguish between single and double phosphorylated TEY motifs in CDKL5, that *E. coli* purified CDKL5ΔC is extremely hypophosphorylated, and that such a state cannot be rescued by incubation with catalytically active CDKL5 variants.

### 2.8. Effect of M10V Mutation on E. coli Expression of CDKL5ΔC and Catalytic Activity 

In *P. haloplanktis* TAC125, an M10V substitution helped the production of flCDKL5 by overcoming an internal translational initiation [25], but no information about the impact of such a mutation on the enzymatic activity is available. However, most of the mutations in the catalytic domain of this enzyme are predicted to be disruptive [35]. To investigate such an aspect, Ph_Sumo-Tat-CDKL5(1-352)-His wt and Ph_Sumo-Tat-CDKL5(1-352)-His M10V_KD (kinase dead) were prepared, and their activity compared to the M10V mutant. Figure 8 shows the expression profiles of wt, M10V, and M10V_KD constructs in *E. coli* BL21(DE3). As in the case of flCDKL5 in *P. haloplanktis* TAC125 [25], the wild-type protein suffers from fragmentation which is absent in the M10V mutants (compare lane 4 with lanes 2 and 3 in Figure 8A). When enriched with IMAC, the wt protein is contaminated with a low molecular weight band that is absent in M10V and M10V_KD constructs (compare lane 2 with lanes 1 and 4 in Figure 8B). In our previous work, N-terminal Edman sequencing revealed that this truncated product starts with MNXF with an indetermination in the third position [25]. However, such a sequence is compatible with a start in the M10 position, as expected.

To assess the possible impact of the M10V mutation on CDKL5 activity, the autophosphorylation levels of Y171 in the activation loop were measured. Figure 9A represents the total CDKL5ΔC detected with an anti-His antibody, while Figure 9B shows levels of autophosphorylation with a specific antibody. As summarized in Figure 9C, although Ph_Sumo-Tat-CDKL5(1-352)-His M10V is active, its phosphorylation levels are about 2.5 times lower than the wild-type enzyme, both with and without preincubation with ATP. As expected, the M10V_KD control only presented a very weak—probably non-specific—phosphorylation signal (Figure 9B).

To define the impact of the M10V substitution on inter-molecular phosphorylation, wt, M10V, and KD_M10V protein variants were incubated with EB2, and their phosphorylation levels were analyzed. A typical experiment is represented in Figure 10, where total CDKL5ΔC and EB2 proteins were contextually detected with an anti-His antibody (Figure 10A), while phosphorylated EB2 was measured with an anti-EB2 pSer222 antibody (Figure 10B). As expected, the KD construct did not phosphorylate EB2 (lane 1 in Figure 10A,B), while the M10V variant was less active than CDKL5ΔC wt (lanes 2 and 3, respectively). As summarized in Figure 10C, increasing times of incubation did not lead to increased EB2 phosphorylation levels and demonstrated that CDKL5ΔC M10V is three times less active than CDKL5ΔC wt, in agreement with the autophosphorylation assay.

### 2.9. Resolution of the Internal Translation Start with a Silent Mutation 

Since the M10V mutation made CDKL5ΔC hypoactive, alternative approaches were used to purify the wild-type protein and avoid its contamination with its N-terminally truncated form. First, we moved the His-tag from the C-terminal extremity to the N-terminal end of the wt construct. Although this approach allowed a higher purity of the final preparations after IMAC, mild contamination with the C-terminal fragment was still visible despite the absence of the His tag, suggesting an oligomerization of the full-length protein with its fragment (Appendix A).

Hence, we tried to diminish the internal translational start by introducing a silent mutation that hampers mRNA-rRNA hybridization. To do so, we first used the RBS calculator to predict the translation initiation rate from the M10-encoding ATG in the CDKL5ΔC transcript [36]. According to this tool, which applies a thermodynamic model to simulate translation initiation, the translation rate at M10 is 195 au which is quite low, suggesting that such an internal translational start is cryptic and difficult to predict. Then, we modeled the effect of each silent mutation in the eight codons upstream of the target ATG on the internal translation rate. As shown in Figure 11A, most nucleotide substitutions would increase such a parameter. However, two mutations at the level of the G7 encoding triplet are predicted to be extremely destabilizing. Following this rationale, we generated a new CDKL5ΔC variant named H6-*CDKL5ΔC* wt_G7opt harboring a silent GGT-GGG mutation at the level of the G7 residue, which is predicted to reduce the internal translation rate from 195 au to 23 au. Figure 11B shows that such a new variant (lane 3) was considerably less fragmented than the original wt one (lane 1) but did not completely abolish the internal translation as the M10V mutant (lane 2), as indicated by the arrow. Furthermore, the application of the same solution produced a similar result for flCDKL5 produced in *P. haloplanktis* TAC125, indicating the validity of the generalization of this approach to different prokaryotes (Figure 11C).

### 2.10. Elimination of Hsp60 from CDKL5ΔC Preparations 

The two main contaminants of CDKL5ΔC wt preparations were its N-terminally truncated product (Figure 8) and Hsp60 when the expression was performed in BL21(DE3, Figure 4). The first issue was solved by moving the Hist tag to the N-terminal end of the protein (Appendix A) and hampering an internal translational start with a silent mutation (Figure 11). As for Hsp60, it has been suggested to be a typical contaminant in affinity chromatography, where Hsp60 coelution with the target protein may be due to either direct interaction with the protein of interest or the chromatographic support [37]. In the first case, the chaperone could be removed by incubation with MgATP in the presence of K^+^ ions [38], while in the latter ionic strength and imidazole concentration in the washing buffer may play a pivotal role [37].

In our experimental setup, it is clear that Hsp60 at least partially interacts with the chromatographic resin, given that it was recovered in the IMAC elution fractions even in the absence of CDKL5ΔC (Appendix A). However, to test both hypotheses we performed three IMACs with different intermediate washes: imidazole wash; low salt wash; and K^+^ MgATP wash.

As shown in Figure 12A, the imidazole wash allowed for the disposal of most of Hsp60 (lane 6), while the low salt buffer was not useful in this regard. On the other hand, the K^+^ MgATP wash removed Hsp60 entirely. However, the SDS-PAGE analysis of more concentrated elution fractions (Figure 12B) indicated that CDKL5ΔC was still slightly contaminated with Hsp60 after the imidazole wash (lanes 1 and 2), it was heavily contaminated both by Hsp60 and other proteins when the low salt buffer was used (lanes 3 and 4), while the K^+^ MgATP wash removed just Hsp60 but not several other contaminants (lanes 5 and 6).

These results suggest that Hsp60 at least partially interacts with the target protein. To define if its release has any impact on the CDKL5ΔC enzymatic activity, a typical EB2 phosphorylation assay was performed (Figure 12C) using either the enzyme recovered after the imidazole wash or the K^+^ MgATP wash as catalyzers (lanes 1 and 2, respectively). Similar amounts of proteins were used for each assay, as shown in the Coomassie-stained gel of Figure 12C (left panel), but the protein kinase released by Hsp60 after the incubation with ATP showed considerably higher activity, as indicated by the anti-pSer222 EB Western blot (right panel in Figure 12C).

These results indicate that both the imidazole and the K^+^ MgATP washes are needed to collect purer fractions of CDKL5ΔC and that the latter might be pivotal to overcome the hypoactivity typical of this kinase when harvested from *E. coli* lysates.

## 3. Discussion

CDKL5 deficiency disorder is a severe neurodevelopmental syndrome caused by mutations in the X-linked *CDKL5* gene [12]. Characterized by the appearance of drug-resistant epilepsy in the very first weeks of life, a plethora of other symptoms follows, such as autistic behavior, locomotor issues, visual impairments, digestive disturbances, and many more [26], which negatively impact the quality of life of affected subjects [39]. Every kind of mutation has been reported, with almost all missense mutations concentrated in the first 350 amino acids, accounting for the protein kinase domain [12,13]. This uneven distribution of point mutations supports the hypothesis that CDKL5 kinase activity is essential for normal brain development. However, the lack of an efficient manufacturing process for CDKL5 production hampered any attempt to understand the structural/functional consequences of CDD mutations. 

In this paper, we embarked on the optimization of the CDKL5 catalytic domain (CDKL5ΔC) production in *E. coli*, with the ambition of supplying the scientific community with a platform useful to study this essential human kinase and possibly screen for chemical compounds able to stabilize or to restore the normal kinase activity of CDD mutants. In line with these aims, we set up a CDKL5-specific in vitro kinase assay, using as substrate the human end-binding protein 2 (*h*EB2), produced and purified from recombinant *E. coli* cells, and the anti-phospho-Ser 222 *h*EB2 antibody to follow the CDKL5-specific *h*EB2 phosphorylation by Western blotting. By this functional assay, we succeeded in testing the quality of different CDKL5ΔC preparations, opening the way to the biochemical characterization of patient-derived mutations on the physiological CDKL5 kinase activity.

The CDKL5 catalytic domain (CDKL5ΔC) has been prepared from different cellular systems, including bacteria [27,28,29], yeasts [2], and insect cells [2,33] to achieve both functional and structural studies. However, no systematic reports about the methodological complications for its preparation are available at present. With this work, we show that when expressed in *E. coli*, CDKL5ΔC tends to aggregate and only a small percentage of soluble and active protein can be collected from the lysate supernatant. Regardless of the application of the most common expedients suggested for the expression of eukaryotic kinases in bacteria, the resulting protein was mostly insoluble, and its soluble fraction tended to associate and co-purify with bacterial chaperones. This feature is common to other difficult-to-express kinases, as frequently reported by other authors [40,41,42], and may be suggestive of the incomplete achievement of their proper folding. The overexpression of such chaperones, as suggested in the past for CDKL5ΔC [27,28,29], was unsuccessful because it just allowed the stabilization of soluble inactive folding intermediates. On the other hand, the fusion of the protein with an N-terminal solubility tag, the recombinant expression at low temperatures, the codon optimization of its gene to avoid an internal translational start, and the addition of K^+^ MgATP was a strategy that overall led to the isolation of a purer and active protein.

Nevertheless, Western blot and mass spectrometry analyses showed that almost the totality of the *E. coli*-derived CDKL5ΔC is not phosphorylated in its activation loop (i.e., the TEY motif in the 169–171 amino acid range) and that only pY171 could be barely detected (0.8%). Surprisingly, the same protein purified from insect cells was also singly phosphorylated in Y171 (20%) with no phosphorylation detected for T169. This feature may suggest that CDKL5 does not automatically fully phosphorylate its TEY motif regardless of the cellular source and that its hyperphosphorylation is stimulated by external signals, as recently demonstrated in renal cells [43]. This cascade activation, which is typical of MAPK, has also been observed in the RCK kinase MAK that is capable of a weak basal autophosphorylation in its Y of the TDY motif, which is increased only after the cross-phosphorylation of its T in the same motif by CCRK [30]. The crosstalk between CDKL5, CCRK, and MAK has been systematically proven in *Chlamydomonas* and *Caenorhabditis elegans* [6,7,8] and the possibility of CDKL5 phosphorylation by CCRK should also be tested in humans to have an insight in its activation mechanism.

However, we cannot exclude that CDKL5ΔC hypophosphorylation is due to the folding issues faced throughout its expression and purification, or the other way around, that the protein’s incapability to completely autophosphorylate itself is the reason for its instability. For crystallographic studies, CDKL5ΔC has been isolated from Sf9 cells in its phosphomimetic state (T169D, Y171E) with the supplementation of a kinase inhibitor [2,33]. Since our future goal is to study the functional and structural consequences of CDD mutations, this is not a pursuable strategy due to the constitutive catalytic inactivity of the DEE mutant in the activation loop.

Eukaryotic protein kinase domains are often intrinsically unstable as suggested by the fact that in eukaryotic cells they are exclusive clients of the Hsp90-Cdc37 chaperone couple whose counterpart in bacteria is missing [44,45,46]. In this regard, it has been suggested that the kinase domains that are poorly produced in *E. coli* are more likely to be among the top interactors of the Hsp90 system [45]. Interestingly, in a recent proximity labeling experiment in primary neurons, CDKL5 was shown to statistically significantly interact with Hsp90 [47], although other studies should be carried out to decipher how strong this interaction is and if it has an important role in influencing CDKL5 stability and regulation.

## 4. Materials and Methods

### 4.1. Cloning Procedures

All the primers used in this project are listed in Appendix A.

Ec_Tat-*CDKL5*(1-352)-His was obtained with PCR amplification using pET28-*CDKL5* as a template (WO2018005617A2 patent, kindly provided by Dr. E. Ciani, University of Bologna) and EcTat11CDKL5_NdeI_fw and EcCDKL5dC_XhoI_rv primers. Then, it was cloned into pET40b plasmid using NdeI/XhoI double digestion.

For Ph_Sumo-Tat-CDKL5(1-303)-His M10V, Ph_Sumo-Tat-CDKL5(1-352)-His M10V, and Ph_Sumo-Tat-CDKL5(1-498)-His M10V, the pB40-BEC-107 (L) M10V plasmid was used as a template [25] and the forward primer was PhSumoCDKL5_NdeI_fw. The reverse primers were PhCDKL5_303_XhoI_rv, PhCDKL5_352_XhoI_rv, and PhCDKL5_498_XhoI_rv, respectively. All these constructs were cloned into pET40b with NdeI/XhoI double digestion.

To generate the untagged Ph_CDKL5(1-352)-His M10V encoding gene, we used PhCDKL5_NdeI_fw forward primer together with PhCDKL5_352_XhoI_rv primer and cloned the amplicon into pET40b with NdeI/XhoI double digestion.

To produce an N-terminally His-tagged version of CDKL5ΔC we amplified its encoding sequence with PhSumoCDKL5_NdeI_fw and PhCDKL5_352_XhoI_rv primers and cloned into pET28a digested with NdeI and XhoI in frame with the His-tag encoding sequence.

GST-tagged versions of CDKL5ΔC, GST-PhCDKL5(303), GST-PhCDKL5(352), and GST-PhCDKL5(498) were achieved using the same reverse primers listed for Sumo-tagged versions, and PhCDKL5_2_BamHI_fw as a forward primer. Then, each gene was cloned into pGEX4T1 with BamHI/XhoI double digestion.

All kinase-dead (KD) variants of PhCDKL5ΔC were obtained using pB40-BEC-107 (L) KD plasmid as a template harboring M10V, K42R, and K43R mutations [25]. Other site-specific mutations were introduced into the already cloned CDKL5ΔC encoding genes using the QuikChange II XL kit (Agilent Technologies, Santa Clara, CA, USA). In particular to revert the M10V mutation and repristinate the wt sequence of PhCDKL5ΔC we used PhCDKL5_V10M_fw and PhCDKL5_V10M_rv primers. To produce the H6-CDKL5ΔC wt_G7opt variant in which the internal translation start is suppressed through a silent GGT-GGG mutation, we used PhCDKL5_G7_GGT_GGG_wt and PhCDKL5_G7_GGT_GGG_rv primers.

### 4.2. Expression of CDKL5ΔC Variants in E. coli Strains

All CDKL5ΔC constructs were either expressed from pET40b (either untagged or Sumo-tagged) or pGEX4T1 (GST-tagged) plasmids using either 50 µg/mL kanamycin or 100 µg/mL ampicillin for selection, respectively. CDKL5ΔC production was tested in BL21(DE3), Shuffle T7, and ArcticExpress(DE3) *E. coli* strains. Cultures of ArcticExpress(DE3) required further selection with 20 µg/mL gentamicin. CDKL5ΔC co-expression with *E. coli* molecular chaperones was achieved in BL21(DE3) using the Takara chaperone plasmid set (#3340, Takara Bio Inc., Shiga, Japan), including pG-KJE8, pG-Tf2, and pTf16 plasmids. They were all selected with 20 µg/mL chloramphenicol and induced at the start of the cultures with either 0.5 mg/mL arabinose or 5 ng/mL tetracycline or both, as indicated by the provider.

Expression trials were performed in LB and M9 supplemented with 0.3% *w*/*v* glucose at 37, 25, and 15 °C, with a 0.1–1 mM IPTG concentration range. Medium composition and inducer concentration did not significantly affect protein production and solubility, while a temperature of 15 °C was needed to increase CDKL5ΔC solubility.

Hence all cultures were carried out by culturing *E. coli* at 30 °C until 0.6 OD, when induction was triggered with 0.4 mM IPTG and a shift of the temperature to 15 °C for 20 h. Then, the cells were harvested by centrifugation (4500× *g*, 20 min, 4 °C), washed with PBS, re-centrifuged, and stored at −80 °C.

### 4.3. Analysis of the Solubility of CDKL5ΔC Variants 

1 OD_600 nm_ *E. coli* pellets were thawed and resuspended in 70 µL of 30 mM sodium phosphate pH 7.0, 0.5 M NaCl, 1% Triton, 1 mM DTT, 5 mM MgCl_2_, 50 U/mL DNAseI, 2 mg/mL lysozyme, complete EDTA-free protease inhibitor cocktail (Merck KGaA, Darmstadt, Germany). After a fast freeze and thaw cycle at −80 °C, the suspensions were incubated at 25 °C for 30 min and then centrifuged (14,000× *g*, 45 min, 4 °C). After separation, the soluble fraction was recovered and denatured with 23 µL Laemmli buffer 4x, while the insoluble pellet was treated with 93 µL Laemmli buffer 1x at 90 °C for 20 min. A 3 µL amount of each denatured fraction was analyzed via 10% SDS-PAGE in TGS buffer after Coomassie staining.

### 4.4. CDKL5ΔC Variants Purification via Native IMAC

For purification purposes, frozen cellular pellets deriving from a 50 mL culture (~200 OD) were used. The cellular paste was resuspended in 13 mL binding buffer (25 mM Hepes pH 7.7, 15 mM imidazole, 0.7 M NaCl, 1% Triton X-100) supplemented with 10 mg/mL lysozyme, and complete EDTA-free protease inhibitor cocktail (Roche, Basel, Switzerland). Then, the suspensions were frozen with dry ice for 10 min and thawed in 25 °C water. Afterward, the suspension was mixed on a tilting plate at RT (25 °C) for 30 min. The crude lysate was then centrifuged for 40 min at 4 °C at the maximum speed of a tabletop Eppendorf centrifuge. After centrifugation (14,000× *g*, 45 min, 4 °C) and 0.2 µm filtration, the supernatant was loaded onto a gravity column packed with 0.6 mL Ni Sepharose 6 Fast Flow resin (Cytiva, Washington, DC, USA). After an extensive wash with binding buffer, most of the contaminants were removed with wash buffer containing an increased imidazole concentration, and a lowered Triton X-100 amount (150 mM imidazole and 0.01% Triton X-100). Finally, CDKL5ΔC was collected by using elution buffer (25 mM Hepes pH 7.5, 500 mM imidazole, 0.5 M NaCl, 0.01% Triton X-100). Imidazole was removed using diafiltration (Vivaspin 6, PES, 10 kDa, Sartorius) with three washes with binding buffer depleted of imidazole. Finally, glycerol was added to 10 *v/v* and CDKL5ΔC aliquots were preserved at −80 °C.

The protocol for the on-column removal of Hsp60 was modified from [41]. In this case, after the binding, the 0.60 mL Ni Sepharose 6 Fast Flow resin (Cytiva) was split into 0.2 mL aliquots that were poured into three separate columns. One column was washed with the wash buffer reported above to remove the quota of Hsp60 interacting with Ni^2+^, the second was treated with low salt buffer (25 mM Hepes pH 7.9, 0.4 M NaCl, 0.01% Triton X-100, 5% *v/v* glycerol, 0.5 mM DTT) to test if Hsp60 hydrophobically interacts with the resin, the third was incubated with the K^+^ ATP buffer (25 mM Hepes pH 7.9, 0.4 M KCl, 10 mM MgCl_2_, 5 mM ATP, 0.01% Triton X-100) to wash away the CDKL5ΔC-bound Hsp60. After 20 column volumes of each wash step, the columns were incubated in the same buffers for 1 h at room temperature (25 °C), before elution as reported above. It is important to point out that the different CDKL5ΔC preparations obtained, albeit characterized by quite different purification levels (where the imidazole-washed sample reached the highest purity), display different enzymatic activities, and the protein eluted by K^+^ MgATP wash was much more active (Figure 12, panel C, lanes KMA). As receiving a biologically active preparation of the CDKL5ΔC fragment is one of the main objectives of this work, we decided to keep the lower level of purification reached in the sample eluted by K^+^ MgATP wash.

### 4.5. Purification of Ec_Tat-CDKL5(1-352)-His from E. coli Aggregates

BL21(DE3) cells harvested after Ec_Tat-CDKL5(1-352)-His expression at 15 °C were lysed as reported in the previous section. After centrifugation (14,000× *g*, 4 °C, 45 min), the cellular pellet was extensively washed with 30 mL cold water four times with intermediate centrifugation steps. A 10 mg amount of washed inclusion bodies (Ibs) were solubilized in 0.5 mL of the buffers indicated in Appendix A at 4 °C for 16 h under mild agitation. After centrifugation (14,000× *g*, 30 min, 4 °C), the soluble fractions were dropwise diluted in 4.5 mL of a refolding buffer containing 50 mM sodium phosphate pH 7.3, 0.5 M NaCl, 5% *v/v* glycerol, 10% *w*/*v* mannitol, 0.01% Triton X-100, 10 mM MgCl_2_, 1 mM DTT and incubated with mild agitation at 4 °C for 3 h. After centrifugation (14,000× *g*, 30 min, 4 °C), soluble protein preparations were checked for purity (SDS-PAGE), quantity (Bradford assay, Bio-Rad, Hercules, CA., USA), and catalytic activity (EB2 kinase assay).

### 4.6. EB2 Recombinant Production and Purification

EB2 was produced in *E. coli* BL21(DE3) bearing the pET40b-eb2 vector encoding for full-length human EB2 [25]. The recombinant cells were grown in LB medium in the presence of 50 μg/mL of kanamycin at 37 °C. At 0.5 OD/mL, 0.1 mM IPTG was supplemented, and the temperature was shifted to 15 °C overnight. Then the bacterial cells were harvested by centrifugation at 10,000× *g* for 30 min at 4 °C.

To purify the recombinant protein, a 500 OD cell pellet was resuspended in 20 mL lysis 50 mM Tris-HCl pH 8.0, 0.5 M NaCl, 20 mM imidazole, 5% *v/v* glycerol, 10 mM MgCl_2_, 50 U/mL DNAse I supplemented with Complete EDTA-free protease inhibitor cocktail (Roche). The suspension was subjected to disruption by sonication using an MS72 probe, 20% Amplitude for 30 min with 30 s pauses (Sonopuls Ultrasonic Homogenisers HD, Bandelin electronic GmbH & Co., Berlin, Germany). After centrifugation at 14,000× *g* for 45 min at 4 °C, the clarified lysate was recovered and applied to a 1 mL HisTrap HP column (GE Healthcare, Chicago, IL, USA) using the Akta Pure FPLC purifier system (GE Healthcare). Then, the column was washed with 50 mM Tris-HCl pH8.0, 500 mM NaCl, and 5% *v/v* glycerol with increasing imidazole concentrations (50 mM and 100 mM, respectively), and the elution of bound proteins was performed with 50 mM Tris-HCl pH 8.0, 500 mM NaCl, 500 mM imidazole, 5% *v/v* glycerol. The eluted fractions were pooled and subjected to a size exclusion chromatography (Superdex 75 prep grade, GE Healthcare) performed using 50 mM Tris-HCl pH 8.0, 100 mM NaCl, and 5% *v/v* glycerol as the running buffer. About 500 μg of purified protein was collected from a 1 L culture. It was aliquoted at 40 µg/mL and stored at −20 °C.

### 4.7. In Vitro Kinase Assays

To monitor the autophosphorylation capability of CDKL5ΔC variants on Thr169 and Tyr171, 200 nM proteins were incubated at 30 °C for 2 h in 20 mM Tris-HCl pH 7.7, 0.5 M NaCl, 10 mM MgCl_2_, 1 mM DTT, 0.7 mM ATP, complete protease inhibitor cocktail (Roche) and Halt phosphatase inhibitor cocktail (Thermo Fisher Scientific, Waltham, MA, USA). Volumes equivalent to 150 ng of CDKL5ΔC were denatured with Laemmli buffer 4x at 70 °C for 20 min and analyzed via SDS-PAGE, and anti-His, anti-CDKL5 pTyr171, and anti-CDKL5 pThr169/pTyr171 Western blots. Levels of autophosphorylation were obtained by dividing the intensity of phosphorylation signals for total protein content (measured either by Coomassie staining or anti-His Western blots). For the cross-phosphorylation assays, equimolar amounts (200 nM) of proteins were mixed and a CDKL5ΔC variant purified from Sf9 cells (#ab131695, Abcam, Cambridge, UK) was used as a positive control.

EB2 phosphorylation assays with pure CDKL5 proteins were carried out using 200 nM EB2 and 100 nM of enzyme in 30 µL of the same buffer as for the autophosphorylation assays at 30 °C. If not differently specified, the reactions were stopped after 30 min with 10 µL Laemmli buffer 4x and denatured at 70 °C for 20 min. A 10 µL amount of each reaction was analyzed via Western blot with either an anti-His (for total CDKL5 and EB2 detection), anti-EB2 (for total EB2 detection), or anti-EB2 pSer222 antibody (for phosphorylated EB2 detection). Levels of cross-phosphorylation were obtained by dividing the intensity of anti-EB2 pSer222 signals for total EB2 and CDKL5 signals.

### 4.8. Western Blot Analyses

After SDS-PAGE runs, proteins were electroblotted to PVDF membranes using a semidry system. After the incubation with specific antibodies, the chemiluminescent signals were developed with the ECL method (Bio-Rad).

In the case of anti-His Western blots, the membrane was blocked with PBS, 5% *w*/*v* milk for one hour. Then, monoclonal anti-polyhistidine-peroxidase clone HIS-1 antibody (#A7058, Merck) was 1:2000 diluted in PBS, 0.05% Tween 20, 5% *w*/*v* milk. After one hour of incubation at RT with the antibody, the membrane was washed with PBS, 0.05% Tween 20 three times (5 min each) and it was developed.

For anti-Flag detection, the membrane was blocked with PBS, 0.2% Tween 20, and 5% *w*/*v* milk for one hour. Then, monoclonal ANTI-FLAG^®^ M2, Clone M2 (#F1804, Sigma-Aldrich, St. Louis, MO, USA) was 1:1000 diluted in the same buffer. After overnight incubation at 4 °C with the primary antibody, the membrane was washed with PBS, 0.2% Tween 20 three times (5 min each) and incubated with an anti-mouse antibody 1:5000 diluted in PBS, 0.2% Tween 20, 5% *w*/*v* milk for one hour at RT. Then, the membrane was washed again with PBS, 0.2% Tween 20 three times (5 min each), and the secondary antibody was detected using the ECL method.

For EB2 detection, the membrane was blocked with TBST (20 mM Tris, 137 mM NaCl, 0.1% Tween-20 pH 7.6), 5% *w*/*v* milk for one hour. Then, the MAPRE2/EB2 antibody (#00109234, Covalab UK Ltd., Cambridge, UK) was 1:4000 diluted in the same buffer. After overnight incubation at 4 °C with the primary antibody, the membrane was washed with TBST three times (5 min each) and incubated with an anti-mouse antibody (#NA931, GE Healthcare) 1:10,000 diluted in PBS, 0.05% *v/v* Triton X-100, 5% *w*/*v* milk for one hour at RT. Then, the membrane was washed again with PBS, 0.05% *v/v* Triton X-100 three times (5 min each) and was developed.

To measure EB2 phosphorylation or Ser222, the membrane was blocked with TBST, 5% *w*/*v* milk for one hour. Then, anti-EB2 pS222 antibody (#00117739 Covalab) was 1:4000 diluted in the same buffer. After overnight incubation at 4 °C with the primary antibody, the membrane was washed with TBST three times (5 min each) and incubated with an anti-rabbit (#7074, Cell Signaling Technology, Danvers, MA, USA) antibody 1:2000 diluted in TBST and 5% *w*/*v* milk for one hour at RT. Then, the membrane was washed again with TBST three times (5 min each) and was developed.

For the detection of phosphorylation on Thr169 and Tyr171 of the CDKL5 T-loop, affinity-purified antibodies were purchased from the University of Dundee. They are the anti-CDKL5 phosphoTyr171 antibody (#SA547) and anti-CDKL5 phospho Thr169 Tyr171 (#SA546) antibodies. Before their use, these antibodies were diluted to 1 µg/mL with 10 µg/mL of non-phosphorylated peptides in TBS-T (20 mM Tris-HCl, 137 mM NaCl, 0.2% Tween-20, pH 7.6) with 5% *w*/*v* milk overnight at 4 °C under agitation. This procedure was needed to block nonspecific antibodies raised against non-phosphorylated epitopes. After the transfer, the membrane was blocked with TBS-T containing 5% *w*/*v* milk for one hour. Then, the solution containing the antibody was added for two hours. After three washes with TBS-T (5 min each), an anti-sheep secondary antibody (#A130-101P, Bethyl Laboratories, Montgomery, TX, USA) was 1:10,000 diluted in TBS-T and incubated with the membrane for one hour. After three washes, the membrane was developed.

Concerning the detection of flCDKL5, the membrane was blocked with PBS, 0.05% *v/v* Triton X-100, and 5% *w*/*v* milk for one hour. Then, CDKL5 (D-12): sc-376314 (Santa Cruz Biotechnology, Dallas, TX, USA) antibody was diluted 1:1000 in the same buffer. After one hour of incubation at RT with the primary antibody, the membrane was washed with PBS, 0.05% *v/v* Triton X-100 three times (5 min each) and incubated with an anti-mouse antibody diluted 1:10000 in PBS, 0.05% *v/v* Triton X-100, 5% *w*/*v* milk for one hour at RT. Then, the membrane was washed again with PBS, 0.05% *v/v* Triton X-100 three times (5 min each) and the secondary antibody was detected using the ECL method.

### 4.9. Mass Spectrometry Analysis for Investigation of Recombinant Proteins Phosphorylation State

CDKL5ΔC variants purified from *E. coli* and Baculovirus infected SF9 cells were fractionated on 10% sodium dodecyl sulfate–polyacrylamide gel electrophoresis (SDS-PAGE) and the gel was stained with GelCode™ Blue Safe Protein Stain (ThermoFisher Scientific, Waltham, MA, USA). The bands corresponding to the recombinant proteins were excised from the gel and hydrolyzed with trypsin according to [48,49]. Peptide mixtures were extracted in 0.2% formic acid (HCOOH) (Merck Millipore, Burlington, MA, USA) and acetonitrile (ACN) (Merck Millipore, Burlington, MA, USA) and vacuum dried via a SpeedVac System (Thermo Fisher Scientific, Waltham, MA, USA).

Subdigestion of the tryptic mixture from Baculovirus-infected SF9 was performed with GluC endproteinase (Roche) in an enzyme/substrate ratio of 1/50. The reaction was blocked with 20% trifluoroacetic acid (TFA) in a final concentration of 2% and the mixture vacuum dried via a SpeedVac System. Peptide mixtures were dissolved in 10 μL of 0.2% HCOOH (Merck Millipore, Burlington, MA, USA) and analyzed by nanoLC-MS/MS on an LTQ Orbitrap mass spectrometer coupled to the nanoACQUITY UPLC system (Waters Corporation, Milford, MA, USA). Samples were firstly concentrated onto a C18 capillary reverse-phase pre-column (20 mm, 180 μm, 5 μm) and then fractionated onto a C18 capillary reverse-phase analytical column (250 mm, 75 μm, 1.8 μm) working at a flow rate of 300 nL/min, using a linear two-step gradient. MS/MS analyses were performed in ETD and CID modality using data-dependent acquisition (DDA) mode, after one full MS scan (mass range from 300 to 1800 m/z) 3 and 5 most abundant ions were selected, respectively, for the MS/MS scan events, applying a dynamic exclusion window of 40 s [50,51].

Output data were processed generating mgf files employed for protein identification procedure in an in-house CDKL5 specific database deposited in Mascot licensed software (Matrix Science, Boston, MA, USA version 2.4.0). Protein mapping was carried out by using 10 ppm as peptides mass tolerance for MS and 0.6 Da for MS/MS search: carbamidomethyl (C) as fixed modification and Gln→pyro-Glu (N-term Q), oxidation (M), pyro-carbamidomethyl (N-term C), and phosphorylation as variable modifications.

### 4.10. Statistical Analysis

Activity assays were based on densitometric analysis of Western blot bands using Bio-Rad Image Lab. T-tests were performed using GraphPad Prism 10 with *n* = 3 and setting *p* < 0.05 as the condition for statistical significance.

## 5. Conclusions

This work demonstrates that the catalytic domain of CDKL5 tends to severely aggregate during overproduction in *E. coli* strains. However, through the implementation of various expedients, that constitute a significant improvement of the previous literature, it is possible to achieve a soluble and active protein with basal phosphorylation of its activation loop. Hypophosphorylated CDKL5 will be useful for future comparative stability and structural studies, and to dissect how posttranslational modifications can affect the protein structure, stability, and activity. Furthermore, the setup of an in vitro activity assay using the purified hEB2 protein as CDKL5 substrate represents a milestone towards the functional characterization of either wild-type or pathological CDKL5 mutants. Future fine biochemical studies are needed to characterize the folding and the oligomeric state of the protein after its purification to better distinguish if such hypophosphorylation is physiological or induced by quality issues. We hope that our paper will foster these studies, highly required to shed light on the structural/functional relationships of this elusive, although essential, brain protein kinase.

## Figures and Tables

**Figure 1 ijms-25-08891-f001:**
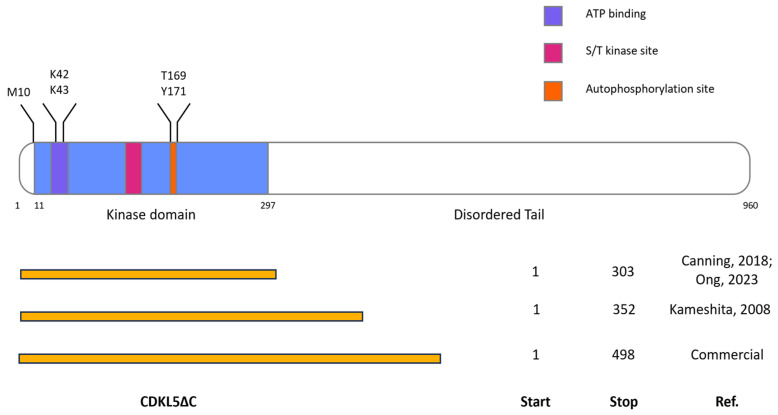
Architecture of full-length CDKL5 (flCDKL5) and CDKL5ΔC variants. The top picture shows the modular structure of flCDKL5 with its functionally relevant parts. In blue the enzyme region corresponding to the kinase domain region (aa 11–297). The highlighted amino acids were either subjected to mutagenesis (M10V to ablate an internal translation start [25], KK42,43RR to generate a catalytically inactive enzyme [27]) or subject to analysis for autophosphorylation (T169 and Y171 of the TEY motif). The bottom scheme shows the boundaries of the different CDKL5ΔC variants tested in this work. Canning, 2018 [2]; Kameshita, 2008 [27]; Ong, 2023 [33].

**Figure 2 ijms-25-08891-f002:**
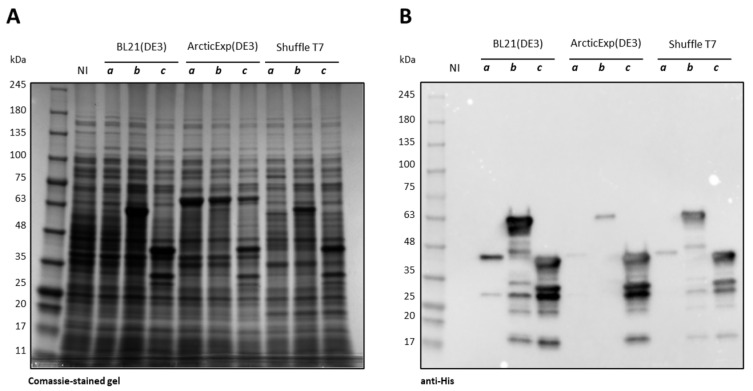
Production of Ph_Tat-CDKL5(1-352)-His M10V (45 kDa) (*a*), Ph_Sumo-Tat-CDKL5(1-352)-His M10V (57 kDa) (*b*), and Ec_Tat-CDKL5(1-352)-His (44 kDa) (*c*) in *E. coli* strains at 15 °C. (**A**) SDS-PAGE and (**B**) anti-His Western blot were carried out on total cellular extracts after 20 h of induction. NI, not-induced BL21(DE3); ArcticExp (DE3), ArcticExpress(DE3).

**Figure 3 ijms-25-08891-f003:**
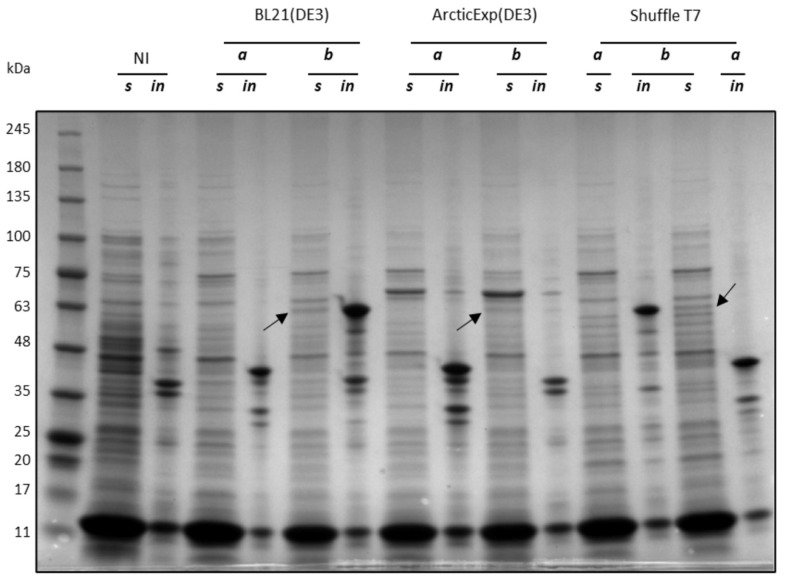
Solubility of Ec_Tat-CDKL5(1-352)-His (44 kDa) (*a*), and Ph_Sumo-Tat-CDKL5(1-352)-His M10V (57 kDa) (*b*) produced in *E. coli* strains at 15 °C. Cellular pellets harvested after 20 h of induction were lysed and soluble (*s*) and insoluble (*in*) fractions were analyzed via SDS-PAGE. NI, not-induced BL21(DE3); ArcticExp (DE3), ArcticExpress(DE3). Arrows point the bands corresponding to the expected recombinant proteins in the soluble extracts.

**Figure 4 ijms-25-08891-f004:**
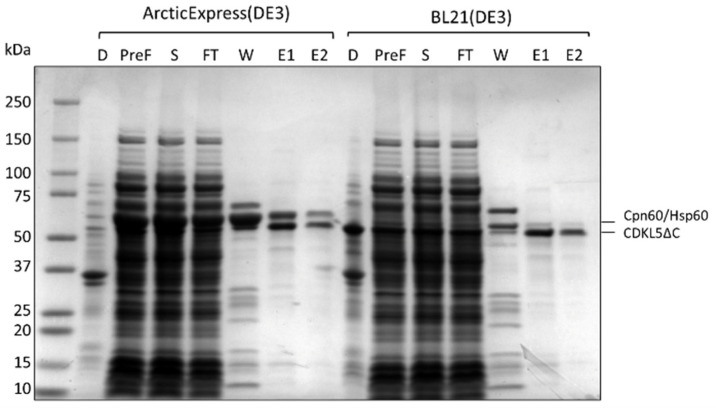
Ph_Sumo-CDKL5(1-352)-His M10V purification from *E. coli* ArcticExpress(DE3) and BL21(DE3) soluble lysates. After cellular disruption, the insoluble (D) and soluble (PreF) fractions were separated by centrifugation. The PreF fraction was further clarified with 0.2 μm filtration (S) and loaded onto an HisTrap FF Crude column. The unbound material was collected (FT) and unspecific binders were removed with an intermediate wash (W). Finally, elution was performed in two steps (E1 and E2).

**Figure 5 ijms-25-08891-f005:**
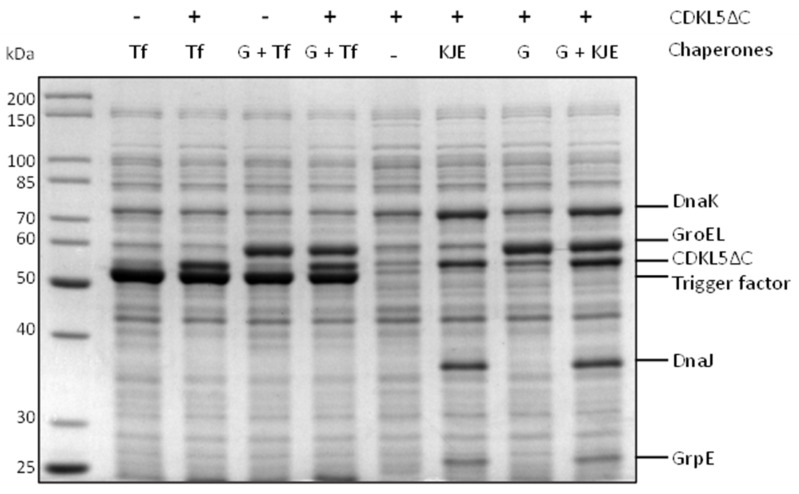
Solubility of Ph_Sumo-CDKL5(1-352)-His M10V in the presence of *E. coli* molecular chaperones. After recombinant production for 22 h at 15 °C, soluble extracts were analyzed via SDS-PAGE. Tf, trigger factor; G, GroELS; G + Tf, GroELS + trigger factor; KJE, DnaK + DnaJ + GrpE; G + KJE, GroELS + DnaK + DnaJ + GrpE.

**Figure 6 ijms-25-08891-f006:**
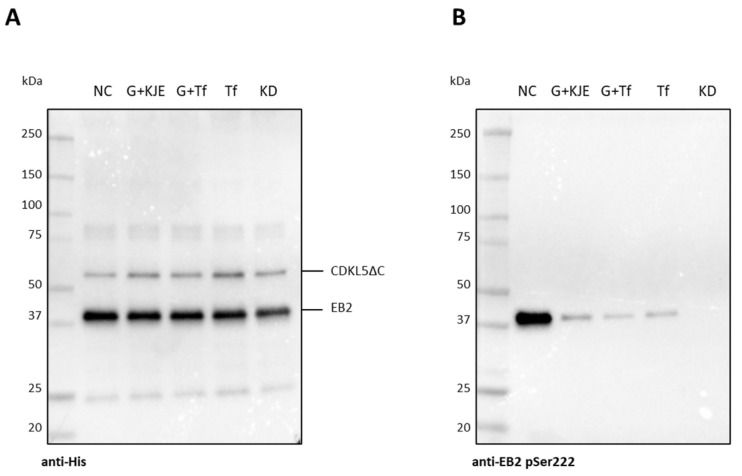
Activity of CDKL5ΔC purified from different *E. coli* strains. (**A**) Anti-His Western blot for the detection of total CDKL5ΔC and EB2. (**B**) Anti-EB2 pSer222 Western blot for the detection of phosphorylated EB2. NC, CDKL5ΔC alone; G + KJE, GroELS + DnaK + DnaJ + GrpE2; G + Tf, GroELS + trigger factor; Tf, trigger factor; KD, CDKL5ΔC KD alone.

**Figure 7 ijms-25-08891-f007:**
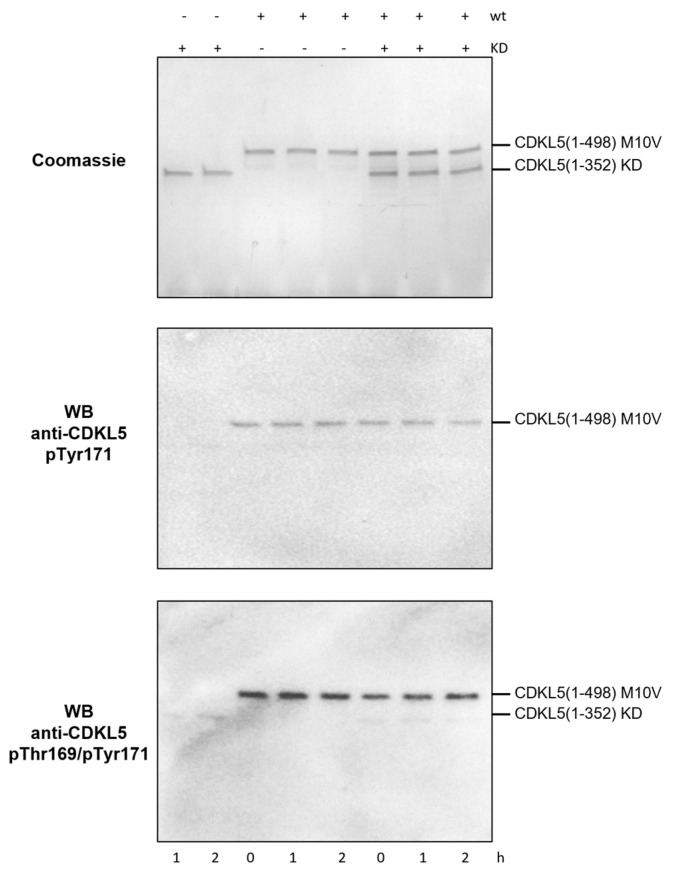
Autophosphorylation of Thr169 and Tyr171 in CDKL5 TEY motif. A catalytically active CDKL5 variant spanning 1-489 residues (Ph_Sumo-Tat-CDKL5(1-498)-His M10V) and a shorter catalytically inactive CDKL5 variant (Ph_Sumo-Tat-CDKL5(1-352)-His M10V_KD) were incubated either together or alone with ATP at 30 °C for 2 h. Representative samples were analyzed at increasing time points via SDS-PAGE (**upper panel**), anti-CDKL5 pTyr171 Western blot (**middle panel**), and anti-CDKL5 pThr169/pTyr171 (**bottom panel**).

**Figure 8 ijms-25-08891-f008:**
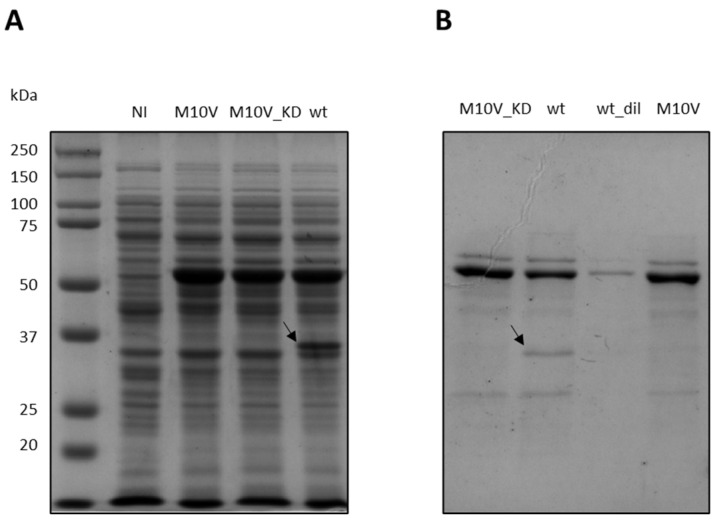
Preparation of Ph_Sumo-Tat-CDKL5(1-352)-His M10V, M10V_KD and wt variants. (**A**) SDS-PAGE of total extracts of BL21(DE3) after recombinant expression. NI, not-induced (**B**) IMAC elution fractions of Ph_Sumo-Tat-CDKL5(1-352)-His proteins. wt_dil, diluted wt. Arrows highlight a Ph_Sumo-Tat-CDKL5(1-352)-His wt fragmentation product due to an internal translation start.

**Figure 9 ijms-25-08891-f009:**
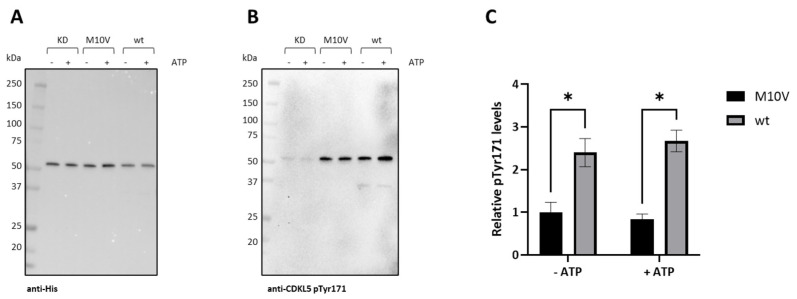
Autophosphorylation levels of Ph_Sumo-Tat-CDKL5(1-352)-His constructs. M10V_KD, M10V, and wt constructs were either incubated alone or in presence of ATP. Then, total CDKL5ΔC was detected with an anti-His antibody after an electroblotting (**A**), while phosphorylated CDKL5ΔC was revealed by an anti-CDKL5 pTyr171 antibody (**B**). Normalized phosphorylation levels were scaled to the M10V construct (**C**). The results are reported as mean of triplicates and the error bars represent standard deviations. Asterisks indicate *p* < 0.05 in Student *t*-tests.

**Figure 10 ijms-25-08891-f010:**
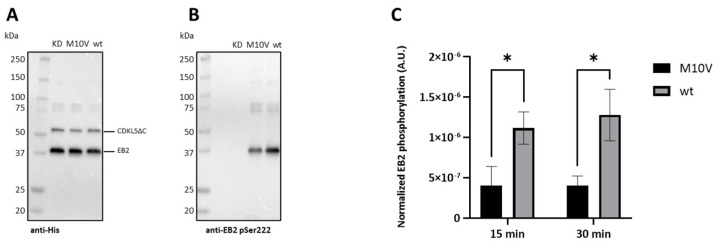
EB2 phosphorylation by Ph_Sumo-Tat-CDKL5(1-352)-His constructs. After the in vitro kinase assay, total EB2, and Ph_Sumo-Tat-CDKL5(1-352)-His variants were detected with an anti-His antibody (**A**), while phosphorylated EB2 was revealed by an anti-EB2 pSer222 antibody (**B**). q1\1\ a triplicate and contains Ph_Sumo-Tat-CDKL5(1-352)-His variants are in the following order: 1, KD; 2, M10V; 3, wt. (**C**) EB2 phosphorylation levels were detected at two time points and normalized for total EB2 and CDKL5ΔC levels. The results are reported as mean of triplicates and the error bars represent standard deviations. Asterisks indicate *p* < 0.05 in Student *t*-tests.

**Figure 11 ijms-25-08891-f011:**
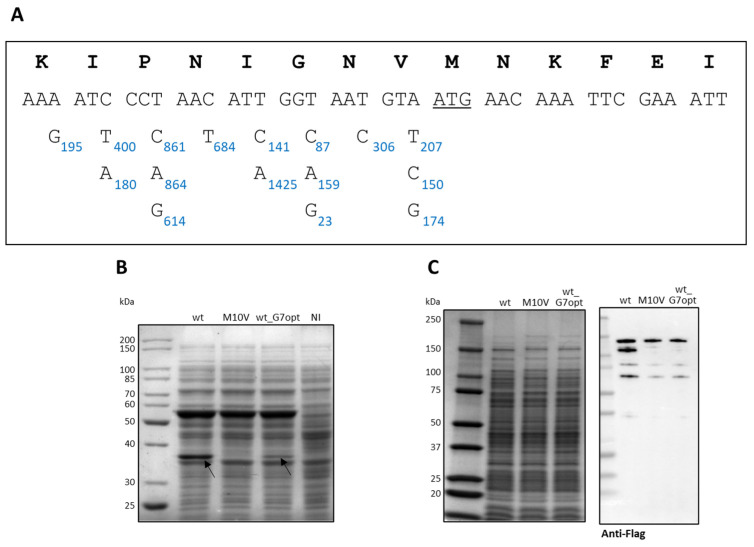
Effects of silent mutations on the internal translation start of CDKL5 encoding transcripts. (**A**) Modeling of the translation initiation rates from the M10-encoding ATG (underlined) in the *CDKL5* transcripts used in this study. The initiation rate of the unmutated transcript was 195 au according to RBS Calculator [36], while the predicted effect of each substitution on such a parameter is indicated by blue numbers. (**B**) SDS-PAGE of total extracts of BL21(DE3) after recombinant expression of CDKL5ΔC variants. NI, not induced. (**C**) SDS-PAGE (left panel) and anti-flag Western blot (right panel) of *P. haloplanktis* TAC125 total extracts after recombinant expression of flCDKL5 variants. The arrows indicate the truncated form of CDKL5ΔC synthesized from M10.

**Figure 12 ijms-25-08891-f012:**
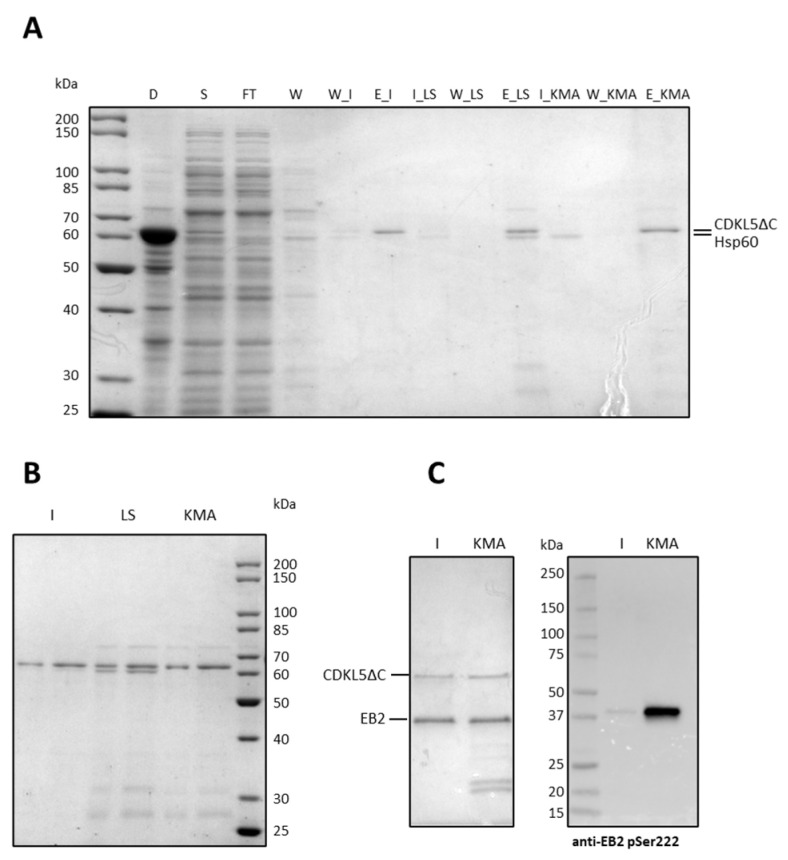
Removal of Hsp60 contamination with on-column washes. (**A**) SDS-PAGE of chromatographic fractions. After cellular lysis, the insoluble fraction was removed by centrifugation (D), and the soluble fraction was loaded onto a Ni-NTA resin by gravity (S). After the flowthrough collection (FT), the resin was washed with 50 mM imidazole (W) and split into three aliquots. The first was washed with the imidazole buffer (W_I) and then the elution was performed with 500 mM imidazole (E_I). The second aliquot was first incubated and then washed with several column volumes of low salt buffer (I_LS and W_LS, respectively), before elution (E_LS). The last chromatography was performed by incubating and washing with K^+^ MgATP (I_KMA and W_KMA, respectively), before elution (E_KMA). (**B**) SDS-PAGE of fractions eluted after imidazole wash (I), low salt wash (LS), and K^+^ MgATP wash (KMA) (1.5 and 3.0 µL, respectively). (**C**) Activity of CDKL5ΔC eluted after either the imidazole wash (I) or the K^+^ MgATP wash (KMA) on EB2. Total proteins were detected by Coomassie staining (left panel), while the product of the reaction was detected with an anti-pSer222 EB2 antibody via Western blot.

**Table 1 ijms-25-08891-t001:** Variables assessed for the recombinant production of CDKL5ΔC in *E. coli*.

Boundaries	Codon Composition	Mutations	Solubility Tags	Chaperones	Expression Temperature	Lysate Fraction
1–303 1–352 1–498	*Escherichia coli* (Ec) *Pseudoalteromonas haloplanktis* TAC125 (Ph) Manual optimization	M10V KK42,43RR	Sumo GST	Cpn60/10 DsbC GroELS DnaK DnaJ GrpE Trigger factor	15–37 °C	Soluble Insoluble

**Table 2 ijms-25-08891-t002:** CDKL5ΔC yields achievable from *E. coli* lysates after co-expression with chaperones. Tf: trigger factor.

Chaperones	Volumetric Yield (mg Protein/L of Culture)
None	6.21
GroELS, DnaK, DnaJ, GrpE	13.75
GroELS, Tf	7.31
Tf	20.6

## Data Availability

All data analyzed during the current study are available from the corresponding author upon reasonable request.

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
