# Peer review of "Bacterial Production of CDKL5 Catalytic Domain: Insights in Aggregation, Internal Translation and Phosphorylation Patterns"

_ijms, 2024, doi:10.3390/ijms25168891_

Round 1
Reviewer 1 Report
Comments and Suggestions for Authors
The authors of the publication “Bacterial production of CDKL5 catalytic domain: Insights in aggregation, internal translation and phosphorylation patterns “ demonstrated a number of detailed studies using various techniques. The author describes the topic in an interesting and understandable way.
Mom, however, has a few comments:
- What antibody did you use as a loading control in the Western Blot method? In order to assess the correctness of the test and the possibility of comparing the results, such antibodies are used to determine the level of a protein, e.g. GAPDH. Do you show something like this?
- In the description of the drawings (e.g. Figure 2), two "a" symbols are used for two different things, which complicates the reception. Especially since you use a capital “A” in the description and there is no such mark in the figure (only small “a”). This should be corrected everywhere so that the markings do not overlap.
- No statistical significance in bar charts (Figure 9,10)
- I also don't see the statistics subchapter in the methodology, what tests were used? This needs to be completed.
- Mass spectrometry was performed in the study, but apart from the information in the additional materials and the description of the method, there is no more information in the publication. e.g. why this analysis was performed and what was obtained as a result. I don't see it. If so, please indicate where.
- Too few references to other literature in the discussion, which results in poor literature. Literature up to 5 years old constitutes only about 25% of the items. Moreover, the literature is not edited in accordance with the journal's requirements.
- What is the usefulness of this research? This could be developed in a discussion or conclusion.
Author Response
Q1.1: What antibody did you use as a loading control in the Western Blot method? In order to assess the correctness of the test and the possibility of comparing the results, such antibodies are used to determine the level of a protein, e.g. GAPDH. Do you show something like this?
R: Most of the Western blots presented are related to pure proteins whose quantification was performed via either Bradford assay (when purity > 95%) or via Coomassie staining with a BSA calibration curve. The evenness of the loading is appreciable in the Western blot itself developed with the antibody for the target protein (e.g. in Fig. 6a, the anti-His Western blot serves to estimate the quantities of pure CDKL5 and EB2 mixed in each reaction).
In the case of protein quantification in cellular extracts (e.g. Fig. 2) the evenness of the loading was conducted by Coomassie staining of the cellular lysates which is an acceptable method both for bacterial and eukaryotic cells (e. g. https://molecularautism.biomedcentral.com/articles/10.1186/s13229-024-00601-9)
Q1.2: In the description of the drawings (e.g. Figure 2), two "a" symbols are used for two different things, which complicates the reception. Especially since you use a capital “A” in the description and there is no such mark in the figure (only small “a”). This should be corrected everywhere so that the markings do not overlap.
R: we thank the reviewer for this observation, and we corrected the format all over the manuscript.
Q1.3a: No statistical significance in bar charts (Figure 9,10)
Q1.3b: I also don't see the statistics subchapter in the methodology, what tests were used? This needs to be completed.
R: Statistical analysis has been added, and the methods for statistical analysis have been added.
Q1.4: Mass spectrometry was performed in the study, but apart from the information in the additional materials and the description of the method, there is no more information in the publication. e.g. why this analysis was performed and what was obtained as a result. I don't see it. If so, please indicate where.
R: As the enzymatic activity exerted by many other MAP-like kinases is directly related to the phosphorylation status of two residues in the so called activation loop, a method to quantitatively evaluate the phosphorylation of Thr 169 and Tyr 171 in the CDKL5 catalytic domain fragments produced in bacterial and insect cells was established. For the sake of clarity, we introduced a subheading (2.5) in the Result section.
Q1.5: Too few references to other literature in the discussion, which results in poor literature. Literature up to 5 years old constitutes only about 25% of the items. Moreover, the literature is not edited in accordance with the journal's requirements.
R: The discussion was revised and deepened, to meet your and other reviewer’s comments. Furthermore, the issues imposed by the heterologous production and purification of CDKL5 hampered any attempt to characterize its biochemical features, and the literature is therefore very limited and diluted in time. We hope that our paper will foster these studies, highly required to shed light on the structural/functional relationships of this elusive although essential brain kinase.
At the submission stage, the journal does not have specific format requirements and accepts Free Format Submission with the following guidelines for the references: “Your references may be in any style, provided that you use the consistent formatting throughout. It is essential to include author(s) name(s), journal or book title, article, or chapter title (where required), year of publication, volume and issue (where appropriate) and pagination. DOI numbers (Digital Object Identifier) are not mandatory but highly encouraged. The bibliography software package EndNote, Zotero, Mendeley, Reference Manager are recommended.” All the required information has been provided.”
Q1.6: What is the usefulness of this research? This could be developed in a discussion or conclusion.
R: We thank the reviewer as the discussion in its original version does not help the readers to understand the main goal of our research. We added lines 455-469 and 524-526 to the Discussion and Conclusion, respectively. We hope that after this modification the aims and relevance of our research will be now clear.
Reviewer 2 Report
Comments and Suggestions for Authors
Reviewer:
In this study, the author determined the expression of the CDKL5 catalytic domain in Escherichia coli through strategies such as using solubility tags, lowering the expression temperature, optimizing codons, and incubating with K+ and MgATP yield a soluble, to achieve a pure active kinase. The bacterial-expressed CDKL5 shows hypo-phosphorylation compared to the eukaryotic version, indicating that bacteria can produce nearly unmodified CDKL5. This work provides a platform for biophysical comparisons between bacterial and eukaryotic proteins, enhancing our understanding of neurodevelopmental disorders related to CDKL5 dysfunction. While the data are overall presented, the paper written is very well, there are several critical points for the author’s conclusion, which need the author’s attention.
Specific points:
1. There is insufficient function study of CDKL5 in vitro and in vivo. The author needs to design more experiments to identify the function.
2. For the purification of CDKL5, much more non-specific binds showed on the membrane. The author needs to optimize conditions to purify this protein.
3. The labeling is not consistent with the figures. Such as some labeling is capitalized, but some is not.
Author Response
Q2.1. There is insufficient function study of CDKL5 in vitro and in vivo. The author needs to design more experiments to identify the function.
R: This work does not aim to study the function of CDKL5 but to analytically study the protein when expressed in a prokaryotic system to define its advantages and pitfalls in comparison to a eukaryotic system.
Q2.2 For the purification of CDKL5, much more non-specific binds showed on the membrane. The author needs to optimize conditions to purify this protein.
R: As observed by the reviewer, the CDKL5DC fragment (1-352) posed several issues for the optimization of IMAC chromatography conditions. As shown in figure 12, several different on-column washing-treatments were explored, to achieve a better purification/activity compromise. Indeed, although the protein eluted by imidazole wash is apparently cleaner (Fig. 12, panel B, lanes I), the protein eluted by K + MgATP wash is much more active (Fig. 12, panel C, lanes KMA). As getting a biologically active preparation of the CDKL5DC fragment is one of the main objectives of this work, we decided to take the level of purification obtained. It is also important to highlight that the in vitro kinase activity on EB2 is CDKL5-specific and in any way related to the presence of the residual protein contaminants, as the kinase-dead mutant preparation (figure 10, panel A and B) does not display any EB2 kinase activity.
Q2.3. The labeling is not consistent with the figures. Such as some labeling is capitalized, but some is not.
R: we thank the reviewer for this observation, and we corrected the format all over the manuscript.
Reviewer 3 Report
Comments and Suggestions for Authors
Dear Authors, your article is very interesting, it is well-written and belongs to the aims and scopes of the journal, presents some novelty but needs more…
Discussion needs to be deeper.
Introduction is too long
Insert docking to explore their interactions
Where is the Mass Spectrometry data? Please, insert them.
Future perspectives
How your work is useful to individuals? Insert this!
Strategies to solve/ attenuate this
Others:
Lines 217-219: eliminate
The article is not in the template of the Journal, please remove the space between paragraphs. Insert paragraphs in the initial of each sentence.
Revise everything.
“Kameshita et al. [27] needed the co-expression of GroELS with CDKL5ΔC in E. coli, we”
Insert the numeration of subsections
also have
Kinase-dead

Author Response
Q3.1 Discussion needs to be deeper.
R: We thank the reviewer as the discussion in its original version does not help the readers to understand the main goal of our research. We added lines 455-469 and 524-526 to the Discussion and Conclusion, respectively. We hope that after this modification the aims and relevance of our research will be now clear.
Q3.2 Introduction is too long.
R: in the revised version of the manuscript, lines 52-53 and 68-69 were removed. We do hope that after these changes the introduction will be more understandable and straightforward.
Q3.3 Insert docking to explore their interactions.
R: If the reviewer is asking us to apply in silico techniques to study the interaction between the CDKL5 catalytic domain and its substrate, human EB2, although a very interesting task, it is outside the scope of the present paper. We set up the phosphorylation in vitro assay (based on the commercial availability of a phospho-Ab which specifically recognizes CDKL5-dependent EB2 phosphorylation site) to evaluate (in a semi-quantitative fashion) the enzymatic activity of the different CDKL5 fragments produced in different fermentation processes.
Q3.4 Where is the Mass Spectrometry data? Please, insert them.
R: For the sake of clarity, we introduced a subheading (2.5) in the Result section.
Q3.5 Future perspectives. How your work is useful to individuals? Insert this!
R: We thank the reviewer as the discussion in its original version does not help the readers to understand the main goal of our research. We added lines 455-469 and 524-526 to the Discussion and Conclusion, respectively. We hope that after this modification the aims and relevance of our research will be now clear.
Q3.6 Strategies to solve/ attenuate this.
R: We are sorry, but this indication is not clear.
Others:
Q3.7 Lines 217-219: eliminate
R: We don’t agree with the reviewer, as the sentence introduces the main results reported in figure 5, and does not represent a repetition.
Q3.8 The article is not in the template of the Journal, please remove the space between paragraphs. Insert paragraphs in the initial of each sentence. Revise everything.
R: At the submission stage, the journal does not have specific format requirements and accepts Free Format Submission with the following guidelines for the references: “Your references may be in any style, provided that you use the consistent formatting throughout. It is essential to include author(s) name(s), journal or book title, article or chapter title (where required), year of publication, volume and issue (where appropriate) and pagination. DOI numbers (Digital Object Identifier) are not mandatory but highly encouraged. The bibliography software package EndNote, Zotero, Mendeley, Reference Manager are recommended.” All the required information has been provided.”
Q3. 9 “Kameshita et al. [27] needed the co-expression of GroELS with CDKL5ΔC in E. coli, we”
R: The citation was placed in the position suggested by the reviewer.
Q3. 10 Insert the numeration of subsections.
R: Done
Round 2
Reviewer 2 Report
Comments and Suggestions for Authors
I suggested that the author do some function tests for CDKL5.
Author Response
Comment 1) I do agree with the authors on this point. The authors did study the activity of the purified construct including its autophosphorylation, which may be considered as functional assays using the setup of the manuscript. Therefore, more functional assay are not essential for the conclusions drawn, and might be out of the scope of the article. However, the function of the kinase is indeed an important point and should be better discusseed in the manuscript.
R: We are in debt with you and the reviewer 2 as the comments raised allowed us to better describe the impact of our results. In particular, we added a sentence in the Discussion (lines 470-475) to address the novelty and importance of the establishment of an in vitro activity assay for CDKL5, based on he use of a recombinant purified CDKL5 substrate (hEB2) and a specific anti phosphoSer222 antibody to follow the CDL5 specific phosphorylation by Western blotting.
Comment 2) I do agree that the manuscript can benefit from a better purification of the kinase, but at this stage of the review and since two other reviewers were satisfied from it, I believe that it is not really necessary to justify the conclusions of the article. It is suggested to explain these points in the Material and Methods sections.
R: Following your suggestion, we introduced a sentence in the Material and Methods section (lines 608-615) to clarify that the less pure catalytic domain preparation (obtained by the elution following the K + MgATP wash) was kept as final purification standard as it turned out in the in vitro assay (see above) to be more active than the purest preparation (obtained by the imidazole wash).
Reviewer 3 Report
Comments and Suggestions for Authors
Thank you for your improvements.
Author Response

(The authors gave the same response as above.)
